# Implementing evidence ecosystems in the public health service: Development of a framework for designing tailored training programs

**Laura Arnold**[1,2]*, **Simon Bimczok**[1], **Timo Clemens**[2], **Helmut Brand**[2], **Dagmar Starke**[1], **on behalf of the EvidenzÖGD study consortium**[¶]

**1** Academy of Public Health Services, Duesseldorf, Germany, **2** Department of International Health, Care and Public Health Research Institute—CAPHRI, Faculty of Health, Medicine and Life Sciences, Maastricht University, Maastricht, The Netherlands

¶ Membership of the EvidenzÖGD study consortium is provided in the acknowledgments.
* arnold@akademie-oegw.de

**Data Availability Statement:** All relevant data are included in the paper and in the supporting information files. To ensure data protection,

## Abstract

The COVID-19 pandemic has highlighted the importance of local evidence ecosystems in which academia and practice in the Public Health Service (PHS) are interconnected. However, appropriate organizational structures and well-trained staff are lacking and evidence use in local public health decision-making has to be integrated into training programs in Germany. To address this issue, we developed a framework incorporating a toolbox to conceptualize training programs designed to qualify public health professionals for working at the interface between academia and practice. We conducted a scoping review of training programs, key-informant interviews with public health experts, and a multi-professional stakeholder workshop and triangulated their output. The resulting toolbox consists of four core elements, encompassing 15 parameters: (1) content-related aspects, (2) context-related aspects, (3) aspects relevant for determining the training format, and (4) aspects relevant for consolidation and further development. Guiding questions with examples supports the application of the toolbox. Additionally, we introduced a how-to-use guidance to streamline the creation of new training programs, fostering knowledge transfer at the academia-practice interface, equipping public health researchers and practitioners with relevant skills for needs-based PHS research. By promoting collaborative training development across institutions, our approach encourages cross-institutional cooperation, enhances evidence utilization, and enables efficient resource allocation. This collaborative effort in developing training programs within local evidence ecosystems not only strengthens the scientific and practical impact but also lays a foundation for implementing complex public health measures effectively at the local level.

excerpts from the qualitative interviews are provided anonymously

**Funding:** The project is part of the funding priority "Strengthening Collaboration between Public Health Services and Public Health Research" of the Federal Ministry of Health and was funded by the federal government Germany.

**Competing interests:** The authors have declared that no competing interests exist.

## 1. Introduction

The systematic incorporation of research evidence into health policy is crucial for the provision of high-quality public health services. Public health professionals are increasingly required to engage in evidence-informed decision-making (EIDM) [1], which aims to identify the most effective and cost-efficient interventions, while also minimizing harm, assessing adverse consequences, and enhancing health outcomes for individuals and communities [2, 3]. Accomplishing this requires that public health policy and practice are informed by the best available evidence, as well as a comprehensive consideration of normative and contextual factors, including political and societal preferences, setting-specific considerations, feasibility, affordability, and sustainability [4–6]. Proper implementation and execution of EIDM has the potential not only to enhance the effectiveness of health policies and public health interventions but also to promote transparency, acceptability, and accountability [7, 8].

A system that encompasses the formal and informal connections and interactions among various stakeholders involved in the production, implementation, and utilization of evidence can be referred to as evidence ecosystem [9]. The formulation and adoption of effective policies and programs relies on the presence of a functional evidence ecosystem encompassing diverse actors from various domains and with diverse agendas [10]. Within such a system, two distinct but interconnected conceptual domains must be considered: evidence generation and evidence utilization, usually linked either to policy or practice [11, 12]. The effectiveness of such an ecosystem relies on robust relationships and active collaboration between public health academia (e.g., universities, research institutes, or academies of public health services) and public health practice (e.g., public health service on federal, state, or local level) [13]. To date, the responsibility of generating evidence has predominantly been ascribed to academic institutions, whereas the utilization of evidence, encompassing the identification, processing, and contextual adaptation of scientific knowledge, has commonly been linked to the domain of public health practice. However, meeting the diverse needs of individuals and communities necessitates a close intertwining of evidence generation and evidence utilization, rather than maintaining a strict separation of responsibilities and competencies. This calls for a symbiotic relationship between academia and practice, enabling the harmonization of scientific inquiry with prevailing practical needs and imperatives [10, 14]. Institutionally anchored, this in turn facilitates informed decision-making in the implementation of evidence-based public health interventions [15, 16].

However, the COVID-19 pandemic highlighted considerable disparities between the envisioned integration of EIDM within a comprehensive evidence ecosystem and the practical implementation of evidence transfer between academia and practice in the German Public Health Service (PHS) [17–19]. In Germany, the health system governance follows a decentralized approach, with responsibilities divided between the federal and state levels, along with corporatist bodies of self-governance [20]. The implementation of federal legislation pertaining to PHS, lies within the purview of the 16 state governments, who wield legislative powers and provide functional and disciplinary oversight of the PHS on local level [21]. Consequently, the local health authorities (LHA) operating at the level of municipalities and independent cities assume a pivotal role in addressing wide range of public health tasks [22]. These LHAs are key actors in promoting and protecting population health and well-being, encompassing health surveillance, evidence gathering, and providing guidance on the prevention of communicable and non-communicable diseases.

Ensuring effective resource utilization and successful intervention implementation requires not only access to scientific knowledge but also its judicious application [6], including the thoughtful integration of evidence into practice, consideration of the specific context, ongoing

evaluation and adaptation of measures and strategies, and continuous stakeholder engagement to ensure ethical, effective, and sustainable outcomes. A judicious approach has the potential to provide decision-makers with better access to information on effective strategies, thereby, optimizing the allocation of limited resources and increasing confidence in the successful implementation of different intervention options [23]. Especially during periods of financial constraints, considering carefully the intricacies of practical implementation within public health decision-making processes safeguards the well-being of the population. This is particularly relevant for the municipal level, as LHAs possess an encompassing understanding of local conditions and contexts predestining them to ensure EIDM processes at the local level [24–26].

Nevertheless, the COVID-19 pandemic has exposed significant challenges in LHAs' implementation and embedding of EIDM processes. These challenges include among others, limited access to scientific information [27, 28], a lack of institutionally anchored collaboration between academia and practice in many places [29, 30], and substantial personnel, material and time constraints that make accessing, reviewing, and applying research findings into local contexts much more difficult [28, 31].

Overcoming these barriers necessitates the establishment of institutionalized evidence ecosystems that foster large-scale collaboration between public health practice and academia and enable inter-agency collaboration in line with the Health in All Policies (HiAP) approach [32, 33]. At the community level, this entails the presence of a well-qualified public health workforce equipped with scientific expertise, administrative skills, and methodological knowledge to ensure a two-way theory-practice transfer.

In Germany, several established training programs provide high-quality education and training opportunities for public health professionals. These programs include, in addition to various public health chairs that primarily prepare students for scientific activities, in particular the Academies of Public Health Services that offer advanced trainings and practice-oriented qualifications. However, to implement and enhance EIDM processes at the academia-practice interface, both comprehensive scientific competencies coupled with extensive expertise in local administrative tasks, activities and responsibilities are needed. This interface encompasses activities aimed at enhancing knowledge transfer between the municipal PHS (e.g., LHA) and research institutions (e.g., universities or academies), as well as fostering collaboration. For brevity, we will refer to this domain as "work at the interface" in the following. Effective dissemination and exchange of evidence at the community level requires a profound understanding of public health services. Consequently, public health professionals working at the interface require competencies in evidence-based policy advice and design, coordination and management skills, and a comprehensive understanding of organizational leadership and decision-making structures [34, 35]. Acquiring these competencies necessitates a consolidation of theory and practice, which is often facilitated through postgraduate training approaches. Noteworthy examples of postgraduate training programs in Germany include the medical residency program for public health specialists [36], the trainee program for public health specialists at LHA Fulda [37], or the postgraduate training in applied epidemiology at the Robert Koch Institute [38]. However, these existing approaches predominantly focus on one occupational group or prepare for single areas of activity. To address this gap, developing training programs tailored to the complex regional characteristics of PHS at the local level is imperative.

The overarching aim of this study was to develop a framework incorporating a toolbox for conceptualizing tailored training programs to equip public health professionals with the necessary skills to enhance evidence-informed approaches at the interface. The study followed a stepwise approach with the following research objectives:

1. Explore the relevant fields of activity, tasks, required skills, and competencies for a position at the interface through several semi-structured key informant interviews (Fig 1: RQ-1/2).

2. Conduct a scoping review to identify, characterize, and analyze available training programs that prepare public health professionals for working at the interface (Fig 1: RQ-3).

3. Develop a toolbox that encompasses key parameters for creating tailored training programs qualifying individuals to facilitate evidence transfer at the local level, drawing from the findings of steps 1 and 2.

## 3. Materials and methods

A stepwise iterative mixed-methods approach was employed, marked by a systematic and cyclical approach in data collection, analysis, and synthesis. This methodology was instrumental in ensuring each phase built upon and refined insights from previous phases for a comprehensive understanding of the topic. Firstly, **qualitative interviews** were conducted to explore the tasks, fields of activities, skills and competencies relevant to working at the interface.

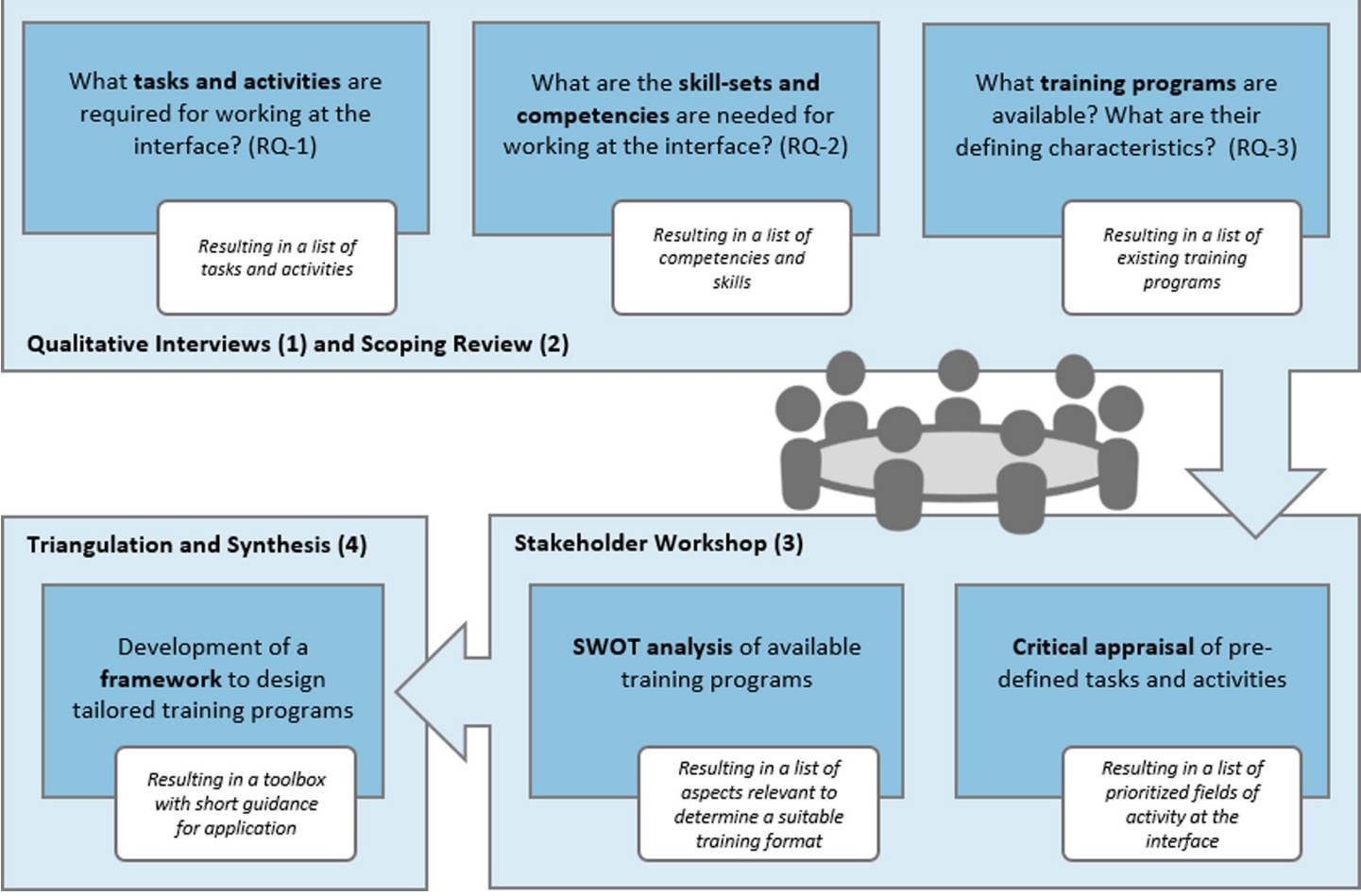

**Fig 1. Schematic of the methodological approach to develop a framework for conceptualizing tailored training programs for working at the interface (RQ = research question).**

Secondly, a **scoping review** was carried out to identify suitable training programs. In the third step, a **stakeholder workshop** was executed, consisting of the critical appraisal and prioritization of the main findings from the previous steps. The final phase, **triangulation and synthesis,** involved a core team of the project consortium reflecting on and synthesizing the results from all previous steps. This iterative process of reflection, appraisal, and synthesis culminated in the development of a comprehensive toolbox that encompasses both essential content and contextual factors required for the development of tailored training programs.

In our study, the terms "framework" and "toolbox" are closely interlinked: The "framework" refers to the overarching theoretical construct guiding the development of tailored training programs. In contrast, the "toolbox" consists of a comprehensive collection of essential elements and parameters, identified during our research. It provides researchers and practitioners with customized guiding questions to identify and discuss all components relevant to their field. Together with its accompanying short application guide, the toolbox is a pivotal aspect of the framework, integral to its function and effectiveness.

The methodological approach is displayed in Fig 1. Details on the activities in each step are described below.

## 3.1 Qualitative interview study

In the first step, 23 semi-structured key informant interviews (KII) were conducted to explore aspects related to public health research, knowledge transfer, collaboration between academia and practice in the PHS, and training programs. Participants were purposefully selected using a sampling plan that utilized professional networks and snowballing. Key informants were carefully chosen based on their expertise in public health academia and practical experience in the PHS. We aimed to achieve a diverse representation, considering a variety of professional backgrounds and levels of experience. Recruitment of study participants took place from December 17, 2021 to March 31, 2022. The first interview was conducted on January 12, 2022, the last on May 03, 2022. During recruitment, all participants were informed about the study procedure and objectives. Written informed consent was obtained from all participants. The consent forms were filed in accordance with German data protection regulations. Prior to recording the interview, all participants were informed about the procedure and verbally asked if they agreed to be recorded. To incorporate the heterogeneity of the PHS accordingly and to reach saturation, the sampling plan considered two dimensions: representation of experts from federal, state, and local governments, and inclusion of experts from public health practice, public health academia, experts with teaching background, and young professionals. A semi-structured interview guide was developed based on existing literature and through brainstorming and iteratively refined by a group of five researchers (LA, DD, SB, SG, and SW) who are all part of the project consortium. After incorporating minor adjustments from the pilot test, the final guide was used for virtual video call interviews. In most cases, participants were invited by e-mail or telephone.

All interviews were conducted by a pair of researchers and the resulting audio files were content-semantic transcribed [39], pseudonymized, and afterwards deleted (LA, DD, SB, SG, and SW). If requested, the transcripts were returned to the participants for correction or comments. No repeat interviews were conducted. The transcripts underwent a deductive-inductive qualitative content analysis following Mayring's approach [40, 41]. After calibration of the coding frame within a group of five researchers from the project consortium, all interviews were coded by two researchers independently using MAXQDA Analytics Pro 2022 (VERBI Software GmbH, Berlin) (LA, DD, SB, SG). Inductive additions to the coding frame were made as

required. Any discrepancies were resolved through discussion. The final coding tree can be found together with anonymized example passages in the appendix (S1 Table).

Relevant tasks, activities, skills, and competencies associated with working at the interface and facilitating evidence transfer at the local level were identified and extracted (RQ-1 and RQ-2, Fig 1). The insights obtained from the interviews were utilized to determine priority areas of activity at the interface. Furthermore, these results informed the subsequent development of the toolbox.

## 3.2 Scoping review of relevant training programs

The second step involved a scoping review aiming to identify, characterize, and analyze available training programs that prepare public health professionals for working at the interface. This methodological approach was chosen due to its comprehensive format, allowing for efficient mapping the existing literature within a limited timeframe while capturing the scope and characteristics of current research activity. We did not attempt to identify all records on training programs at the interface, but enough to assume that saturation had been reached regarding the parameters to be identified in the analysis. To the best of our knowledge, there is currently no such overview.

The scoping review was conducted following the framework proposed by Arksey and O'Malley [42]. To answer RQ-3 (Fig 1), we developed a search strategy focusing on two themes:

- PHS workforce, referring to professionals providing essential public health services within local, state, or national level public health authorities.

- Practice-oriented training programs, such as continuing education, training opportunities, and professional development programs, prepare individuals for academic work in PHS practice or practice-oriented roles in public health academia, including hybrid training programs that bridge both areas.

After piloting and refining the search strategy by two researchers (LA, SB), we searched the scientific databases PubMed and LIVIVO on October 26, 2021 (Appendix: S2 and S3 Tables). Retrieved records were de-duplicated in the bibliographic management software CITAVI (Swiss Academic Software GmbH) and imported in Rayyan, a web-based tool for conducting systematic reviews [43]. Initially, the screening process was calibrated and the predefined inclusion and exclusion criteria were tested for practicability and applicability. Records were assessed for eligibility based on the following criteria:

- *Population*: Included records referred to the PHS workforce as defined above or individuals engaged in academic public health. Excluded records focused primarily on professionals involved in patient treatment.

- *Context*: Included records referred to practice-oriented training and qualification approaches and programs as defined above. Excluded records encompassed training exclusively focused on one area (e.g., new master programs unrelated to the PHS) or lacked the objective of qualifying for the interface.

- *Setting*: Included records focused on training approaches implemented in Germany as well as results from neighboring countries with comparable healthcare systems, including Austria—AT, Switzerland—CH, and the Netherlands—NL. Examples from the United Kingdom—UK, known for its Public Health Specialist Program within the National Health

Service (NHS), were also considered exemplary. Excluded records concentrated on training approaches from other countries.

- *Publication date and language*: The search was restricted to articles published between 2011 and 2021 in English, German, or Dutch.

- *Publication type*: No restrictions were imposed based on publication type.

- *Full-text accessibility*: Excluded records were those for which full-text access was unavailable.

Following the calibration of the screening process, a title-abstract screening and subsequent full-text screening was conducted independently by two researchers (LA, SB). Relevant publications were transferred to MAXQDA Analytics Pro 2022 (VERBI Software GmbH, Berlin) for coding and thematic synthesis analysis based on the approach outlined by Thomas and Harden [44]. A critical appraisal was not conducted in accordance with Arksey and O'Malley's scoping review framework methodology [42]. Initially, three sample documents were coded line-by-line inductively by both researchers, demonstrating a high level of consensus. Subsequently, the remaining documents were coded individually by one researcher each, with new codes developed and added inductively as needed. All coded passages were then grouped, defined, and labeled according to identified similarities and differences, resulting in a hierarchical tree structure (Appendix: S4 Table). From the emerging descriptive themes, a first version of the toolbox was developed (LA, SB).

## 3.3 Stakeholder workshop

The third step involved a multi-professional and interdisciplinary stakeholder workshop, which aimed to critically appraise the first version of the toolbox by prioritizing skill-sets, competencies, and key elements for the development of tailored training programs. The in total 44 participants were purposely selected using a sampling plan similar to the interview approach. The participant pool was intentionally diverse, extending beyond the key informants from the qualitative interviews. It included representatives from various public health organizations, public health practitioners from PHS on local, state and federal level, academic researchers from different universities and research centers, and young professionals. The selection was guided by the intent to include individuals who shared professional and communication skills but offered a range of opinions, to encourage dynamic and critical discussions. This approach aimed to prevent the occurrence of "group think", where decisions are influenced by conformity or dominance of certain individuals within the group [45].

The recruitment phase lasted from April 01, 2022 to June 14, 2022, and the workshop itself took place from July 04–05, 2022.

The workshop utilized the Strategic Orientation Mapping (SOR) approach for the decision-making process proposed by Schlicht and Zinsmeister [45]. Prior to the workshop, participants completed an online survey to prioritize fields of activity at the interface. The results of this survey were presented at the beginning of the workshop. The workshop consisted of two parts: firstly, a critical appraisal of the tasks and activities deemed as priorities in the survey, and secondly, an assessment of purposefully selected training programs using the SWOT analysis approach [46, 47]. The selection of these training programs was a collaborative decision within the consortium, informed by the outcomes of both the qualitative interview study and the scoping review. This selection process was guided by a dual focus: the programs chosen were those deemed most compatible with the operational environment of the PHS workforce, and simultaneously, those that were practice-oriented, facilitating effective work at the interface

between scientific research and practical application. The workshop results were documented, processed by the project team, and shared with participants for feedback and final adaptations.

## 3.4 Triangulation and synthesis

The results from step one to three were combined and consolidated through triangulation in workshops with the research consortium members. Triangulation in this context refers to a systematic integration of results obtained from the various methodological approaches employed in the previous steps [48]. Through an iterative process, data from steps one to three were subjected to reflexive analysis, where the results were compared and synthesized. These consolidated outcomes, summarized by two researchers (LA, SB), were subsequently presented to the project consortium members for critical assessment. The feedback received informed further revisions. This iterative cycle continued until a consensus was achieved. As a result of this cyclical process, a comprehensive toolbox was developed, aimed at facilitating the creation of customized training programs for public health professionals who work at the interface.

Subsequently, this toolbox was utilized in a series of workshops within the project consortium to develop a training program, specifically tailored to the unique structural conditions of all consortium member institutions. A detailed description of the application of the toolbox alongside with the resulting training program can be found in forthcoming publications (Arnold et al., in preparation).

## 3.5 Ethics and reflexive statement

Since the qualitative interviews and the stakeholder workshops contained exclusively technical questions, study-related stresses and risks were expected to be minimal. As some information provided by participants might involve criticism of their own agency or partner organizations, the pseudonymity of data was ensured, and no detailed information was disclosed. The study was conducted in accordance with the Declaration of Helsinki and data collection was approved by the Institutional Ethics Committee of Faculty of Medicine at the Heinrich-Heine-University Düsseldorf, protocol codes 2021–1646 (2021-10-28) and 2021–1646_1 (2021-12-16). We adhered to the Consolidated Criteria for Reporting Qualitative Research (COREQ) checklist in reporting the qualitative interviews, stakeholder workshops, and final triangulation and synthesis [49]. Results of the scoping review were reported according to the Reporting Items for Systematic reviews and Meta-Analyses extension for Scoping Reviews (PRISMA-ScR) checklist [50].

Throughout the qualitative interviews and the stakeholder workshop, our research team maintained active engagement with both the research process and the participants. In line with the COREQ guidelines, we acknowledge that personal biases were not entirely avoidable. This recognition led us to adopt a reflexive stance throughout the research, where we remained aware of our potential biases and their potential impact on data collection and analysis. To mitigate the influence of these biases, we implemented several measures: We critically examined our assumptions, fostering a culture of self-reflection and open discussion. Regular monthly meetings with the project consortium were instrumental in this regard, providing an interdisciplinary platform for evaluating research progress and perspectives.

Our research team brought together a diverse range of expertise, including epidemiology (LA, DD), medicine (DD, HB), public health (LA, SB, HB, TC, SG, DS), sociology (SW, DS), and social sciences (LA, DS). All authors possessed experience in conducting qualitative research projects. This diversity in professional and academic backgrounds played a pivotal role in shaping the research process. During the triangulation and synthesis phase, the multifaceted perspectives within our team proved to be particularly valuable. Each member's

insights and interpretations, shaped by their unique disciplinary lenses, were critical in the development of the final toolbox.

## 4. Results

As a result of the methodology employed, two main outcomes have emerged. The first outcome involves the mapping of relevant areas of activity at the interface. The second outcome pertains to the development of a toolbox designed to create tailored training programs aimed at equipping public health professionals for work at these interface. Both outcomes laid the groundwork for the currently ongoing development and pilot testing of a customized training program tailored to the specific requirements and local needs of the EvidenzÖGD research consortium (Link). A comprehensive report on this aspect will be provided once the piloting phase is completed (Arnold et al., in preparation).

### 4.1 Fields of activity at the interface

A total of 24 individuals were interviewed in 23 semi-structured interviews. Most of the participants had multiple specialties and areas of expertise due to their professional background, which is shown accordingly in Table 1. Of the interviewees, sixteen primarily worked in the PHS and ten at a university. Five participants identified themselves as early career professionals, and eight brought an international perspective.

The duration of the interviews ranged from 31 to 56 minutes. The interviews included an equal representation of men and women. Based on the interviews, areas of knowledge transfer activity at the municipal level were identified and clustered, along with identification of requisite skills and competencies.

The obtained results were further discussed and prioritized in the multi-professional and interdisciplinary stakeholder workshop with 48 experts. Of these, 40 participants responded to the initial question regarding the positioning of their professional expertise in a triangle between practice, academia, and teaching. Among them, 18 experts primarily identified with public health practice, 14 experts primarily with public health academia, and four experts indicated a prior focus on teaching and conceptualization of training programs. Four experts situated themselves at the intersection of all three domains. Additionally, 30 experts reported having five or more years of experience in the field of public health, while 21 experts reported having five or more years of experience specifically in the PHS. Furthermore, 21 experts mentioned their participation in the design of a training program at least once.

The results of the interview analysis and the stakeholder workshop were further elaborated by the interdisciplinary research consortium through a series of conceptual workshops. Finally, the consortium consolidated six fields of activity for establishing evidence-informed processes and structures at the interface between academia and practice in LHAs in Germany:

- *Networking and committee activities*: This field focuses on creating and fostering networking opportunities among stakeholders at the community level. It also includes the mapping of needs, goals, and expectations to promote ongoing engagement.

**Table 1. Characteristics of the interview participants.**

| Current activity in the PHS | Current activity at a university | Early career professionals | International perspective |
|---|---|---|---|
| *Federal level: 9 participants* | *11 participants* | *5 participants* | *8 participants* |
| *State level: 2 participants* | | | |
| *Local level: 5 participants* | | | |

- *Knowledge management (evidence use)*: This field entails building sector-specific and cross-sectoral expertise on municipal PHS ("practical knowledge"). It also includes the development of practice-oriented research questions, the conduction of evidence syntheses including quality assessments, and the identification of suitable implementation approaches.

- *Knowledge communication*: This field involves tailoring expertise and research findings to local conditions by developing appropriate communication materials, such as policy briefs, statements, or evidence synthesis. It also includes the development, implementation, and promotion of a joint communication strategy to facilitate cross-institutional knowledge exchange between stakeholders, team members, leaders, and decision-makers.

- *Project management*: This field encompasses evidence-informed identification, adaptation and subsequent implementation, administration, and evaluation of (complex) public health interventions and measures. It also includes the securement of project funding.

- *Capacity building and change management*: This field focuses on opportunities to strengthen the structural conditions of knowledge-transfer processes, aiming to enhance the capacity of individuals and organizations to apply research evidence effectively.

- *Consolidation of knowledge-transfer processes*: This field involves the sustainable implementation of evidence-informed processes and structures. It also includes establishing feedback mechanisms and promoting individual as well as organizational self-reflection in evidence use.

A detailed description of the knowledge, skills, attitudes, and attributes required for each field can be found in the competency framework developed as part of the EvidenzÖGD project (Link).

## 4.2 Toolbox for the development of relevant training programs

The identification and extraction of relevant aspects for the development of tailored training programs for the PHS workforce on local level was informed by the Six-Step Model for Developing Competency Frameworks proposed by Batt et al. [51]. Emphasis was placed on defining desired outcomes and considering relevant process aspects, including inputs and activities. Continuous evaluation of this process took place throughout the iterative development phase, enabling the utilization of findings to enhance the ongoing processes of model development and revision.

A total of 1,706 records (PubMed: n = 1,468; LIVIVO: n = 238) were identified in the scoping review. Following the title-abstract screening, 1,590 records were excluded, and an additional 24 records were excluded during the full-text screening. Exclusion reasons are displayed in Fig 2. Ultimately, 25 records were included in the review. The characteristics of the included records can be found in Appendix (S5 Table).

Based on these 25 records, four core elements were identified as relevant for the development of a training program intended to qualify professionals for work at the interface: (1) context-related aspects, (2) content-related aspects, (3) aspects relevant for determining the training format and (4) aspects relevant for consolidation and further development of the program. A total of 15 parameters were assigned to these core elements, and specific guiding questions with examples were developed to facilitate the application of the toolbox. Subsequently, each of the four aspects, along with corresponding guiding questions, will be presented.

**4.2.1 Context-related aspects.** **Context-related aspects** involve essential program parameters and factors that need to be established prior to program implementation. These aspects encompass (a) agreement on program objectives, (b) involvement of relevant stakeholders, (c) identification of required resources, and (d) definition of the program setting. To facilitate the

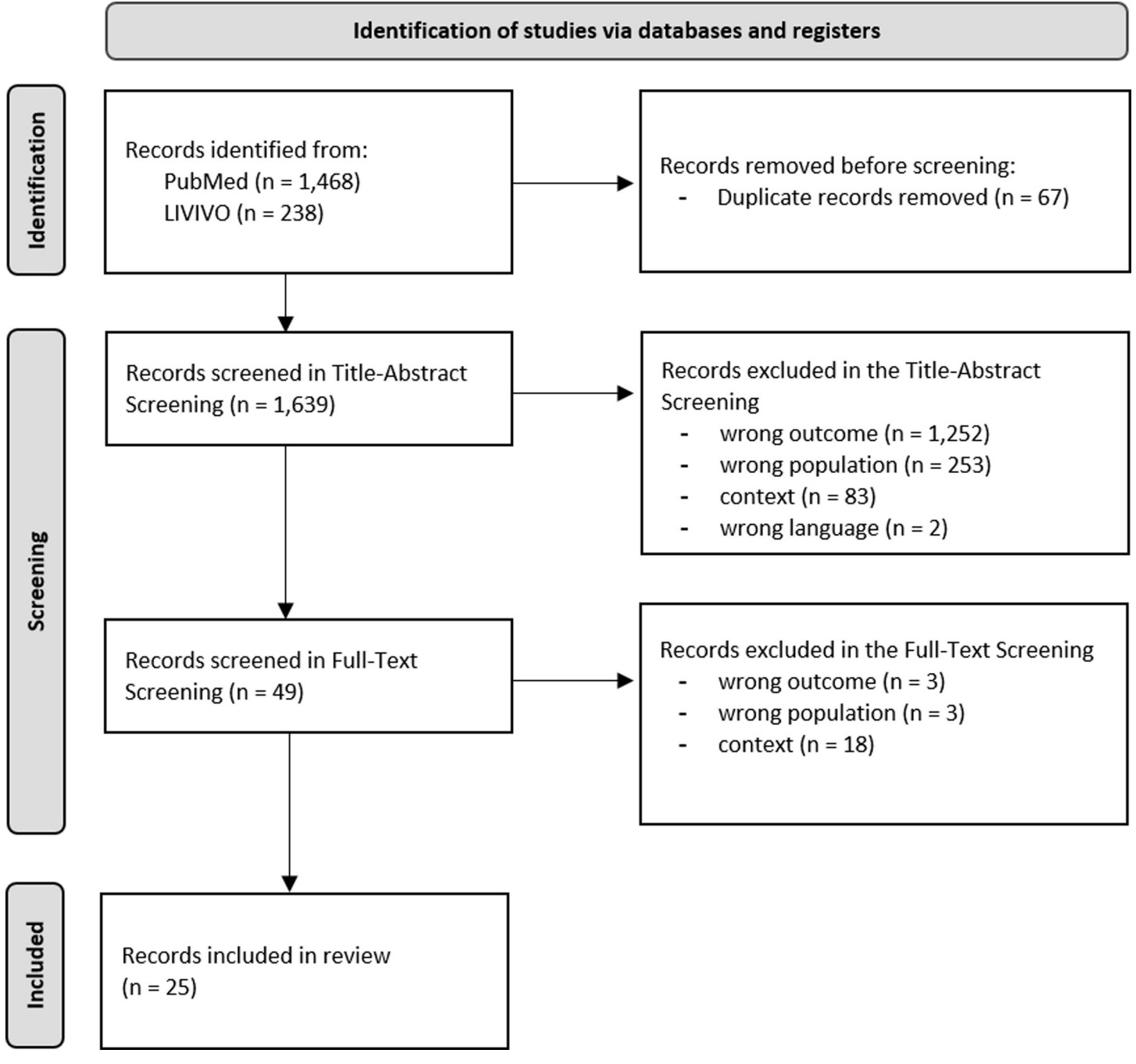

**Fig 2. PRISMA flowchart.**

conceptualization of a training program considering contextual factors, Table 2 presents these parameters and corresponding guiding questions.

**4.2.2 Content-related aspects.** **Content-related aspects** are essential for structuring the program content effectively. These aspects include determining the (a) relevant program content, (b) selecting suitable training and education formats, (c) didactical concepts, and

**Table 2. Context-related aspects for the development of a training program.**

| Parameters | Guiding questions to be answered |
|---|---|
| (a) Program objectives | What are the overarching guiding principles and objectives of the training program? [52–67] |
| (b) Stakeholder involvement | Which stakeholders are involved in the training program?<br>• *Who takes the lead in program management and organization? [52–54, 56–59, 61–66, 68–70]*<br>• *Which stakeholders are included in program administration? [52, 54, 56–59, 61, 62, 64, 65, 67, 68, 69, 71, 72]*<br>• *Who is the target audience of the program? (e.g., Bachelor/Master/PhD students, professionals, specific professions) [52–63, 65–67, 69–71, 73–75]* |
| (c) Necessary resources | What resources are required for program establishment?<br>• *What financial resources are needed for the program? [52–56, 58, 59, 61–63, 66, 67]*<br> • *Is there a financing concept in place for program establishment and maintenance?*<br> • *What are the participation fees for the program?*<br> • *Are participants responsible for covering participation fees?*<br>• *What human resources are needed to deliver program content? [52–54, 57–63, 65, 69, 72, 74, 75]*<br> • *Which qualifications are required for program instructors?*<br>• *What other resources are necessary for program establishment and maintenance (e.g., technical equipment, facilities, literature resources, licenses)? [52, 57, 58, 60–63, 75]* |
| (d) Program setting | In which setting will the program be carried out?<br>• *Where will the program be conducted? (e.g., local, regional, national, international) [65]*<br>• *What setting will be used for the practical component? (e.g., LHA, hospital, general practice) [53, 54, 60, 63]*<br>• *What setting will be used for the theoretical component? (e.g., universities, academies, public health schools) [53, 54, 60, 63]*<br>• *Are there any restrictions to consider? (e.g., distance, daily work routine, internet access) [57, 61, 65]*<br>• *What delivery formats will be used? (e.g., face-to-face, distance learning, hybrid) [52–55, 57–59, 61, 62, 64, 68, 71, 75, 76]*<br>• *In what language will the program content be delivered? [61]*<br>• *Does the program setting ensure equal access participation for all? [57, 61, 65]* |

specifying (d) appropriate measurability and assessment methods. Table 3 illustrates these parameters and provides guiding questions to aid determining the program's content.

**4.2.3 Aspects relevant for determining the training format.** Once the context- and content-related aspects have been established, the next step is to specify the **preferred program format**. Based on the findings of the scoping review, four parameters were identified as relevant for the transition into a training program (Table 4). These parameters include determining the program format (a), program duration and density (b), professional credentialing requirements (c), and the recruitment process and selection strategy (d).

**4.2.4 Program parameters of selected training programs.** Within the scoping review, we identified various qualification models, with five being particularly suitable for facilitating evidence use and knowledge transfer within the local PHS in Germany. These models include trainee programs, PhD programs, rotational concepts, continuing professional development (CPD) courses that integrate both academia and practice, and PHS-related modules in academic degree programs. The selection of these models was based on a collaborative endeavor within the consortium, deeply informed by insights from the qualitative interviews, the scoping review, and the stakeholder workshop. Our criteria for choosing these training programs were twofold: First, we concentrated on programs compatible with the operational environment of the PHS workforce in Germany, ensuring alignment with the specific requirements and challenges faced by professionals. Second, we prioritized programs that were practice-oriented, designed to effectively bridge the gap between scientific research and practical

**Table 3. Content-related aspects for the development of a training program.**

| Parameters | Guiding questions to be answered |
|---|---|
| (a) Program content | **Horizontal Integration**[*1] [54, 65, 72, 74]<br>• Which professions should be integrated in the program's content? [52, 54, 56, 59, 61, 72, 73] (e.g., public health experts, medical doctors)[*2]<br>• Which disciplines should be integrated in the program's content? [54, 56, 59–61, 72–74] (e.g., healthcare, public health, law, ethics)[*2]<br>• How can the curriculum accommodate the diverse professional backgrounds of the public health workforce? [54, 56, 59–61, 72–74]<br>• Should the curriculum allow for individual content preferences? [54, 57, 60, 61, 72, 74, 76]<br>• Should international experience be integrated into the curriculum? [53–55, 57, 61, 62, 76]<br>**Vertical integration**[*3] [52, 59, 60, 62, 65, 68, 74, 76]<br>• What content should be included to enhance knowledge-transfer between practice and academia in the PHS?<br> ○ *Which theories and models should be integrated? [56, 59, 60, 68, 70, 74–76] (e.g., Evidence-based Public Health (EBPH) [6], Essential Public Health Operations (EPHOs) [77], Health in all Policies (HiAP) [78], Information Pyramid [79])*<br> ○ *What content areas should be integrated? [53, 54, 56–65, 67, 68–72, 74–76]*<br> ○ *Should the program have predefined learning outcomes? [53, 54, 58–60, 69, 70, 72, 76] (e.g., WHO-ASPHER Competency Framework [80], UK Public Health Skills and Knowledge Framework (PHSKF) [81], Core competencies in applied infectious disease epidemiology in Europe [82], Catalogue of Learning Objectives Epidemiology [83])*<br>• Should the program follow a competency framework? [53, 54, 56, 57, 59, 61, 64, 67, 69, 70, 74]<br> ○ *If yes, which one? (e.g., European Core Competences for Public Health Professionals [49], Core Competencies for Public Health Professionals [50], Public Health Skills and Knowledge Framework [51], WHO-ASPHER Competency Framework [52], Core competencies in applied infectious disease epidemiology in Europe [53])*<br> ○ *If no, should a new competency framework be developed? (e.g., in accordance with the Six-Step Model for Developing Competency Frameworks [43] and aligned with the CONFERD-HP Guideline [54])*<br>• Which research methods and skills should be delivered in the program? [52, 54, 57, 59–62, 64, 65, 70, 71, 74–76] *(e.g., quantitative and/or qualitative methodology, academic writing, analytical reasoning, critical appraisal, identification of knowledge gaps, problem analysis, understanding scientific language)*<br>• How does the practical application succeed?<br> ○ *How can academia and practice be connected within the content of the program? [52, 54, 57, 59, 60, 65, 70–74, 76] (e.g., by addressing aspects relevant for collaborative relationships between researchers and end users, the involvement of decision makers in research processes, or timely access to research)*<br> ○ *How can the program deliver practical knowledge and address practice-oriented problems? [54, 67, 70] (e.g., by applying problem-oriented or practice-based learning approaches)*<br> ○ *How can the program provide career development options perspectives? [54, 57, 59, 60, 71, 76] (e.g., by providing coaching or mentoring)*<br> ○ *How can the program be relevant to the workplace reality of participants? [54, 58, 60, 61, 63, 71–73, 76] (e.g., by co-creating the qualification model, co-hosting the final training approach)* |
| (b) Training and education forms | Which training and education forms should be applied to deliver the content of the program?<br>• *Which teacher-centered training and education forms should be used?(e.g., (guest) lectures, mentoring, supervision, personal development planning, Q&A formats, train-the-trainer) [52–55, 58–62, 64, 65, 67, 68, 70, 74–76]*<br>• *Which training and education forms promoting interaction and exchange between participants should be employed? (e.g., discussion formats, group exercises, interprofessional knowledge-exchange, journal clubs, networking formats, peer-assisted learning, research projects, tandem models, tutorials) [52, 54, 55, 57–63, 65, 67, 70, 71, 73–76]*<br>• *Which training and education forms emphasizing practical experience for participants should be incorporated? (e.g., internships, networking formats, exposure to practical problems, rotational concepts) [54, 56, 59, 63, 64, 67, 70–72]* |
| (c) Didactical concepts | Which didactical concepts should be used to deliver the content of the program [53–55, 58, 60, 61, 68–74, 76] (e.g., *adult/lifelong learning, problem-based learning (PBL), presentation of real-life problems, research-based learning, self-directed learning, self-reflective learning*)? |
| (d) Measurability and assessment methods | What assessment methods should be used to evaluate participants' understanding of the program's content?<br>• *Should the examination contain a theoretical component? (e.g., exams, presentations, group work formats) [53, 56, 58, 66, 69, 72]*<br>• *Should the examination contain a practical component? (e.g., working on research projects) [56, 59, 62, 71, 72]*<br>• *How can the time spent studying the program's content be made measurable? Should some kind of credit point system be used for this? (e.g., European Credit Transfer and Accumulation System (ECTS) [73, 84])* |

*1 **Horizontal integration** describes the integration of content for different professional disciplines into the curriculum so that participants can adopt a broad public health perspective [74].

*2 We defined "**professions**" as roles obtained through specific training, academic education, or professional trajectory, and "**disciplines**" as overarching fields of work according to [85].

*3 **Vertical integration** describes the linkage of theories, methods, content, and application within the content of the program (bridging the gap between research and practice) [74].

**Table 4. Aspects relevant for determining the training format.**

| Parameters | Guiding questions to be answered |
|---|---|
| (a) Program format | What is the most suitable program format?<br>• *What program forms are generally suitable? [52, 54, 55, 57, 58, 60–64, 67, 68, 69, 72–75] (e.g., hospitations/seminars/courses/workshops, stand-alone vs. integrated into existing programs, postgraduate programs, PhD programs, trainee programs)*<br>• *Can existing program forms be adopted? If not, which core elements seem particularly suitable? To address this question, it is worthwhile to develop an overview of the core elements of purposively selected programs as exemplarily presented in Table 5.* |
| (b) Program duration and density | What is the overall duration of the program?<br>• *What is the most suitable duration of individual modules within the program? [52–56, 59, 61, 62, 64–67, 71, 72]*<br>• *What are the scheduling options for participants? (e.g., full-time, part-time, block sessions, self-determined) [52–54, 56–63, 67, 71, 72]* |
| (c) Professional credentialing requirements | What are the required standards for program completion? [*1]<br>• *Are there professional credentialing standards upon program completion? (e.g., acknowledgements, degrees, certificates, register) [53, 54, 56, 62]*<br>• *What are the accreditation requirements for the program? (e.g., Accreditation body, public health registry, boards) [53, 54, 56, 57, 64, 68]*<br>• *What credentials or certificates do participants receive upon completion? (e.g., degree, certificate, certificate of attendance)? [54, 58, 60, 62]*<br>• *Does program completion grant entry into a (national) public health registry (if existent)? [53, 56, 73]* |
| (d) Recruitment and selection | How are participants recruited and selected?<br>• *What procedure is used to select participants for the program? (e.g., recruitment strategy, written and / or face-to-face assessments, admission exam) [53, 56–59, 62, 71]*<br>• *What selection criteria must be met for program participation? (e.g., required professional background, years of postgraduate experience, completion or enrollment in a Master's program that covers specific areas, appointment to a specific training site) [53, 54, 56, 60, 62]*<br>• *What is the desired number of participants for the program? [53, 56–62, 66]* |

[*1] For more details, see Gershuni et al. [86]. Their systematic review on professional regulation and credentialing of public health workforce contains detailed information on relevant factors to be considered.

application in public health service. This dual-focused selection process was aimed at identifying programs that meet the contextual needs of the PHS workforce while also enhancing knowledge transfer and application at the interface of academia and practice.

Table 5 presents the selected training programs based on their program parameters. All five qualification models were subject of a SWOT analysis during the stakeholder workshop. Special attention was paid to the feasibility of ensuring knowledge transfer at the local level.

**4.2.5 Aspects relevant for consolidation and further development of the training program.** Furthermore, parameters necessary for the long-term existence of the program were classified as **aspects relevant for consolidation and further development of the program**. These aspects encompass a range of factors that can be grouped into three overarching parameters, including (a) piloting and implementation, (b) evaluation and quality assessment, and (c) advancement and transferability. Table 6 provides an overview of these aspects, along with guiding questions that aim to support a comprehensive understanding and careful consideration of the fundamental elements required for ensuring the ongoing success of the program.

**Table 5. Program parameters of selected training programs.**

| | Trainee programs | PhD-programs | Rotational concepts | CPD[*1] courses addressing academia and practice | PHS-modules in academic degree programs |
|---|---|---|---|---|---|
| **(a) Program objectives** | Preparation of qualified (young) professionals for future professional activities in the PHS through a practice-oriented approach. | Investigation of public health-related research problems, addressing the needs of local and state health authorities. Integration of research findings into the practical work of the PHS to facilitate the long-term adoption of evidence-informed approaches in the PHS. | Familiarization of participants with the different working environments of academia and practice in the PHS. Establishment of a strong network between participating institutions. Addressing practical deficits in academia as well theoretical deficits in practice. | Empowering participants to engage in evidence-informed activities related to PHS-specific topics, thereby contributing to quality assurance in daily PHS operations. | Sustainable promotion of scientific knowledge in the public health sector. Facilitation of career orientation for students and young professionals. |
| **(b) Setting** | Practical training in trainee programs frequently takes place across various departments within a LHA or a training institution. Collaboration with (inter)national partners is occasionally involved. | PhD Programs are often based on collaborations between institutions of academia and practice related to public health. PhD students with an academic background often undertake a complementary placement within the PHS. | Rotational concepts typically occur between at least one PHS institution at the municipal or federal level and one academic institution, such as universities, research institutes. Presence in these institutions is obligatory most of the time. | CPD courses addressing academia and practice often encompass a series of individual modules, seminars, or workshops delivered in face-to-face, online, or hybrid formats. | PHS-modules in academic degree programs are typically offered as compulsory or elective subjects within the curricula of undergraduate or postgraduate academic programs, such as public health and medicine. |
| **(c) Optimal duration** | Several months or even years | Custom (usually 3–5 years) | Ranges from permanent positions to limited time durations (e.g., one year). | Ranges from one-day training sessions to multi-year CPD programs. | Custom (often 1–2 semesters) |
| **(d) Core content focus** | Acquiring hands-on skills. Focus on practical and theoretical training components of various PH-/PHS-related topics | Aligning research projects (PhD) with specific practice-based research needs of the PHS. | Learning new skills and methods, applying evidence-based decision-making to practice, and aligning research questions with practical needs. | Acquiring specific skills and/or competencies for working at the interface. Integration of current public health discourses. | Learning about more complex topics, issues, and methods. Variation in module scope, professional depth, and the specific content. |
| **(e) Participant selection criteria** | Academic degrees. Content-related knowledge is sometimes assessed in eligibility tests. | Masters' Degree with thematic relation to the PHS. Practical experience with PHS institutions at the municipal or state level is often beneficial. | Employment in a PHS institution at the municipal or state level or academic institutions, such as universities or research institutes is often required. | Activity in the PHS or another area of the health system is often required. | Dependent on the respective study programs. |
| **(f) additional considerations** | Supervision is often provided by a mentor. | PhD-projects are usually supervised by both university professors and experts from public health institutions, such as LHA. | Encourages participants to integrate their own expertise into the field they are rotating in, benefiting the participating institutions considerably. | - | Ideally, the module includes a diverse range of potential career paths in public health institutions. |
| Examples | • Trainee program for public health specialists in Fulda [37] • Postgraduate Training for Applied Epidemiology (PAE) at the RKI [38] • Public health specialist training [53] | • Doctorate/ Ph.D. Program: Doctor of Public Health (Dr.PH) Doctor of Philosophy (Ph.D. in Public Health) [87] • Ph.D. Program: Medical Research in Epidemiology & Public Health [88] | • Embedded research approaches [52] • Rotation within the medical residency program for public health specialists training in the LHA Hamburg [89] | • Public health literature searching training course within Knowledge and Library Service [75] • Structured Operational Research and Training Initiative [62] | • PHS-relevant courses in the MSc Public Health at the HHU [90] • PHS-relevant courses in the MA Applied Health Science at the RWU [91] |

[*1] CPD = Continuing professional development (CPD).

**Table 6. Aspects relevant for consolidation and further development of the training program.**

| Parameters | Guiding questions to be answered |
|---|---|
| (a) Piloting and implementation | How can a piloting phase before establishing the training program be established?<br>• *How to test feasibility and practicability of the model? [61, 68, 75]* |
| (b) Evaluation and quality assessment | How can the evaluation concept be designed to ensure a comprehensive quality assessment of the program?<br>• *Which components of the program should be evaluated? (e.g., development process, content, performance, acceptability, feasibility, satisfaction)*<br>• *What are the objectives to be achieved through the evaluation process (e.g., assessing program effectiveness, identifying strengths and weaknesses, improving program outcomes, informing decision-making, resource allocation)?*<br>• *Which methods and indicators should be employed to evaluate the program?*<br>  ○ *Level 1: Reaction (e.g., satisfaction with the program and/or the program content, application rate, attendance rate) [60, 61, 63, 64, 66, 71, 72, 76]*<br>  ○ *Level 2: Learning (e.g., graduation rate, assessment of newly acquired competencies, participants' achievements after graduation, employment rate of participants after graduation) [53–55, 57–60, 64, 65, 70–73, 76]*<br>  ○ *Level 3: Behavior (e.g., factors contributing to the successful application of the program content, barriers contributing to the successful application of the program content) [52, 55, 60, 61, 64, 71, 76]*<br>  ○ *Level 4: Results (e.g., impact of the program on practice / on certain institutions / on political debates) [64, 72]*<br>  ○ *Level 5: Return on investment (e.g., cost-effectiveness of the program) [64]* |
| (c) Advancement and transferability | How can the continuity of further program development be ensured?<br>• *Does the program receive sufficient support from relevant stakeholders, such as experts and recognized institutions, to be effective in practice? [56, 69]*<br>• *Should the program be aligned with other existing training programs? [52, 60, 73, 76]*<br>• *How to secure adaptability to developments in the field of public health/in the PHS? [52, 57, 59, 62–64, 68, 69, 71, 72, 75, 76]*<br>• *How can the program be adjusted to address its criticized aspects? [64]*<br>• *What is the log-term financing strategy for the training program?*<br>How can the transferability of the program to other contexts be ensured? [*1]<br>• *Are there good-practice examples of successful transferring the training program to other contexts? [52, 56, 59, 62, 64, 65, 69]*<br>How should the dissemination concept be designed to enhance visibility of the program? [52, 58, 59, 64, 65, 71, 76] |

[*1] For more details, see Schloemer et al. [92]. The authors developed a model for the assessment of transferability of health interventions through identification and systematization of influencing criteria, including facilitators and barriers within a systematic review.

## 5. Discussion

### 5.1 Summary of findings

Current research-oriented training opportunities provided by well-established public health programs effectively prepare students for academic roles but often neglect the complexities of working in and with local governments [93]. Conversely, practice-based training programs often do not adequately manage to equip practitioners with the skills necessary for academic tasks and roles.

   To address the lack of available training offerings that meet the unique needs and requirements of the local context, we have developed a comprehensive toolbox for conceptualizing integrated training programs. These programs aim to enhance evidence transfer between academia and practice in local PHS. Through an iterative process involving key informant interviews, a scoping review, and a multidisciplinary stakeholder workshop, we identified essential aspects and parameters for such a toolbox. The toolbox presented comprises four core elements, encompassing a total of 15 parameters:

1. content-related aspects (incl. four parameters),

2. context-related aspects (incl. four parameters),

3. aspects relevant for determining the training format (incl. four parameters), and

4. aspects relevant for consolidation and further development (incl. three parameters).

Specific guiding questions with illustrative examples have been developed for each parameter to assist in the development of tailored training models aligned with local needs and requirements. These guiding questions enable program developers to effectively assess the complexities associated with developing, implementing, and sustaining the program's effectiveness and impact.

## 5.2 Short guidance on how to apply the toolbox

The toolbox presented is deliberately generic so that it can be applied to different contexts. To this end, the accompanying guiding questions are intended to be supportive to determine the relevant focus and content. While the toolbox can be used standalone by systematically answering all questions in sequence, we believe the toolbox to be most useful in an iterative deliberative process, as recommended by experts in the field [51]. To facilitate this, we propose a seven-step process, visually depicted in Fig 3, with a brief explanation of each step outlined below.

To **initiate the process** (step 1), it is crucial to form a core team comprising representatives from relevant public health institutions on the local level. This core team should include experts with broad professional backgrounds from universities, research institutions, academies of public health services, local public health authorities, local administration, and ideally,

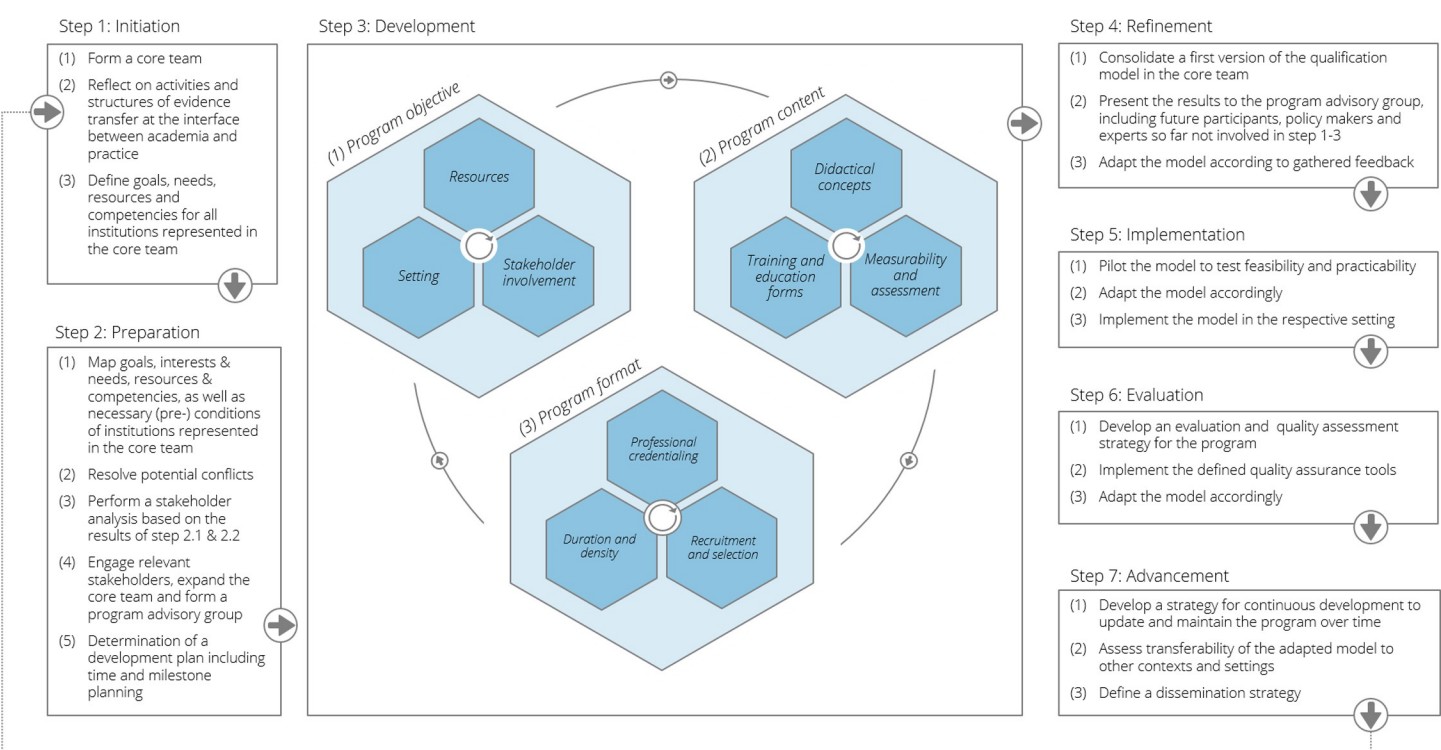

**Fig 3. Schematic how-to-use guidance of the toolbox.**

government agencies. Reflecting on activities and structures relevant for evidence transfer between academia and practice provides insights into existing practices and highlights areas for improvement. This reflection enhances understanding regarding the dissemination of research findings and utilization of evidence in current practice. Ensuring alignment and collaboration throughout the development process is further supported by each entity represented in the core team initially defining their goals, needs, resources, and requirements.

In the **preparation phase** (step 2), the defined goals, interests, needs, resources and requirements should be aligned with underlying competencies of each member institution. This mapping exercise provides a deeper understanding of the strengths and expertise within the core team while identifying any overlaps or conflicts. Promptly resolving potential conflicts enables effective collaboration and smooth progress throughout the development process. Performing a stakeholder analysis based on the results of the mapping will provide insights into the broader ecosystem and helps identifying relevant experts not yet involved. These experts can be engaged by expanding the core team or involving them in the program advisory group, which should be formed to ensure comprehensive guidance throughout the development process.

During the **development phase** (step 3), the core team uses the toolbox to reach a consensus on the fundamental elements of the training program. An iterative process is recommended refining content- as well as context-related aspects by systematically assessing the provided guiding questions (see Tables 2 and 3) in successive workshops conducted by the core team. To ensure continuous exchange within the core team, a predetermined schedule should be agreed upon in the initial meeting. Once program objectives and content are specified, the program format, including duration and density, professional credentialing requirements, and desired recruitment and selection criteria, can be defined (see Tables 4 and 5).

In the **refinement phase** (step 4), the core team consolidates the first version of the qualification model based on the agreed-upon parameters. Subsequently, the model is presented to the program advisory group, including future participants, policy makers, and experts not involved in prior steps. Gathering input and feedback allows for refining and improving the model. The core team adapts the model according to the feedback received, ensuring its relevance, effectiveness, and alignment with stakeholders' expectations.

To test the feasibility and practicability of the model, it is recommended to pilot the model in a real-world setting at the outset the **implementation phase** (step 5) (see Table 6(A)). Careful assessment of results and feedback from the pilot phase enables necessary adaptations to the model. Once refined, the model can be implemented in the respective setting.

During the **evaluation phase** (step 6), efforts should be dedicated to developing a comprehensive evaluation and quality assessment strategy for the program. The core team defines relevant quality assurance tools to assess the program's effectiveness, impact, and adherence to standards (see Table 6(B)). Continuous monitoring and evaluation, accompanied by adjustments, ensure the ongoing quality improvement of the tailored qualification model.

In the final **advancement phase** (step 7), a strategy for continuous development to update and maintain the program over time should be established (see Table 6(C)). Assessing the transferability of the adapted model to other contexts and settings, while considering scalability and applicability, is crucial. Defining a dissemination strategy facilitates sharing the knowledge and experiences gained during the development and implementation process, ensuring broader adoption and impact of the model.

## 5.3 Need for institutional anchored evidence eco-systems on local level

Effective implementation of needs-based public health interventions requires the assessment, synthesis, and appropriate utilization of research evidence, in alignment with the broader

policy system. The practical challenges encountered by LHAs in this regard became evident during the COVID-19 pandemic, where time-sensitive decisions had to be made amidst epistemic uncertainties [5, 17]. This was aggravated by absence of structures facilitating rapid knowledge transfer and exchange in many municipalities, particularly at the pandemics' onset. To bridge the gap between evidence generation and utilization in the local PHS in Germany, the establishment of evidence ecosystems is deemed highly relevant.

Comprehensive methodological skills, including formulating appropriate research questions and conducting evidence syntheses, are essential to obtain timely and robust evidence on public health-related challenges. Successful implementation of complex interventions relies on the effective translation of new research findings from academia into practice and vice versa. Therefore, sector-specific and cross-sectoral expertise in public policy and administration is vital for aligning research questions with local needs, as emphasized by experts in the field [23, 94–96], particularly on the local level. To recommend and implement locally tailored strategies, scientific evidence must be communicated in a meaningful and usable manner for policy-makers, decision-makers, and practitioners [2, 97]. This necessitates the production of succinct and user-friendly evidence syntheses, specifically tailored to meet informational demands of the intended users. Additionally, addressing local needs entails active engagement and involvement of key stakeholders, interdisciplinary teams of experts, and collaborative and continuous efforts between evidence generators and evidence users [10].

The toolbox presented aims to serve as a guide to conceptualize training programs that teach skills related to evidence generation and to train participants in evidence utilization by translating and applying parts of a generic body of evidence to the community context in which LHAs operate. Within the EvidenzÖGD-project we utilized the toolbox by following the proposed step-wise approach of the how-to-use-guide to develop a customized training program tailored to the specific needs of the research consortium. The resulting qualification model is currently undergoing piloting and evaluation (Arnold et al., in preparation).

Tailored training programs are intended to equip public health researchers and practitioners with the relevant skills to design and implement needs-based PHS research. The joint development of training approaches seeks to strengthen cross-institutional collaboration and enhance understanding of evidence generation and utilization. In the long term, this process is meant to enable public health researchers and practitioners to conduct high-quality PHS research, aligned with local needs, thereby saving resources and enhancing the evidence base for successfully conducting complex public health measures on site. Consequently, co-developing tailored solutions within a local evidence ecosystem can contribute not only to scientific impact but, potentially, to practical impact as well.

However, the successful implementation and effectiveness of such tailored programs will also be influenced by contextual conditions. In addition to adequate material and financial resources, laws and regulations that mandate evidence-informed decision-making processes are essential. Therefore, a clear political endorsement and support for the integration of evidence-informed practices into the policy and practice on local level are needed. Moving forward, attention to these contextual factors is vital to ensure the optimal impact and sustainability of tailored training programs.

## 6. Strengths and limitations

The design of this study incorporates some noteworthy strengths. The novelty of our overarching approach, in which we looked at aspects relevant to strengthen knowledge-translation and exchange at the interface of academia and practice in local PHS in Germany by taking into account evidence required from a scoping review, several key-informants and a multi-

professional group of experts. The comprehensive mixed-methods approach allowed us to incorporate a broad variety of methods and types of evidence. The iterative approach enabled critical evaluation of our own research results, contributing to continuous quality assurance and consolidation. Based on this, we assume that our study, which is jointly organized by academia and practice, can contribute significantly to the improvement of knowledge-transfer processes at the municipal level by means of the proposed toolbox and its application.

However, the study possesses some methodological limitations that warrant consideration. Firstly, the search of the scoping review was restricted to five European countries. This decision was primarily made to identify training programs that have been implemented or tested within a context comparable to the German public health system (AT, CH, and NL) or in a well-established context (UK). We are aware of other good examples from the United States and Canada, among others [3, 98–101]. Due to the fact that we supplemented the results with 23 key-informant interviews and finally reflected and prioritized the entire toolbox with 48 experts, we assume relevant coverage. It is important to note, that our aim was not to capture all available records within the scoping review. However, the achieved saturation gives us confidence that the approach captured the majority of relevant publications and reflects on the majority of relevant parameters. Some parameters (e.g., "piloting and implementation" or "advancement and transferability") were underrepresented in the identified training programs, which might be due to the fact that these aspects are so far not regularly considered in training program development.

Secondly, the qualitative interview study had a strong focus on the German context, with limited inclusion of international experiences. Nevertheless, a small sample of well-informed international interviewees provided valuable insights into current debates within their country contexts. While this emphasis on individuals' experiences within the system strengthens the development of tailored training programs, it may limit the emergence of "out of the box" thinking.

Lastly, the multidisciplinary stakeholder workshop allowed for open discussions, benefiting from participants diverse professional backgrounds, ranging from students to retired public health experts. This facilitated the integration of different disciplines and varying levels of expertise and interests. Although all topics were successfully discussed and prioritized as planned, it is worth considering alternative approaches, such as a DELPHI process, to generate additional solutions. However, it should be noted that the time and availability of the experts involved was limited and other approaches would have been much more time-consuming. In this regard, the commenting phase following the stakeholder workshop proved valuable in mitigating potential biases associated with group thinking processes, while also partially addressing the time constraint limitation.

While our study explicitly focused on enhancing knowledge translation and exchange within the decentralized governance structure of the public health system in Germany, we recognize the potential limitations of the toolbox's applicability to systems with different structures. Our interviews and scoping review are tailored to the specific needs of the German context and therefore focus on five selected European countries, which we consider a strength. However, we intentionally designed the toolbox with a generic framework to enable adaptation to various regions and contexts. We acknowledge the potential utility and necessity of this adaptability in designing customized training programs for different systems. The toolbox's transferability to other geographical and administrative contexts is currently being explored in the ongoing EvidenzÖGD study. The extent to which the toolbox can effectively serve in diverse administrative systems and settings without a decentralized governance structure must be examined in the future.

## 7. Conclusions

This study aimed to develop a toolbox that serves as a guide to develop training programs to equip public health professionals and researchers with the knowledge, skills, and capacities relevant to implementing evidence-informed approaches at the interface of academia and practice. Training programs that are explicitly tailored to local needs have the potential to foster a shared research culture focusing on topics relevant for the PHS and establish a sustainable cross-institutional infrastructure known as a local evidence ecosystem.

Applying the toolbox for training program development by following the proposed how-to-use-guide can contribute to the strengthening and enhancement of the local evidence ecosystem in which they are embedded. The efficacy of this approach should and will be evaluated in future studies. If successful, a well-established evidence ecosystem can provide the much-needed bridge between the evidence-generating and the evidence-utilizing system. This mutually beneficial relationship benefits both public health academia and practice, as research questions and projects tailored to local needs can lead to the development of customized solutions. Consequently, an institutionally anchored knowledge-transfer and exchange ecosystem helps to ensure a transparent and evidence-informed fulfillment of local PHS tasks and activities.

## Supporting information

**S1 Table. Coding tree structure and anonymized exemplary passages from the qualitative interviews.**
(XLSX)

**S2 Table. Search strategy for the database PUBMED.**
(DOCX)

**S3 Table. Search strategy for the database LIVIVO.**
(DOCX)

**S4 Table. Overarching coding tree structure from the scoping review.**
(DOCX)

**S5 Table. Characteristics of the studies included in the scoping review.**
(DOCX)

## Acknowledgments

The authors express their gratitude to the study participants for their time and dedication. Additionally, we would like to extend our appreciation to Katharina Kreffter, Lena Raith, Meret Reuther, Joy Pirig, and Luisa Urban for their valuable support throughout the project. We also thank Jan M. Stratil for his valuable time and constructive feedback on earlier manuscript versions.

In this study, the contributions of the EvidenzÖGD study project consortium were integral to the research progress. The consortium—a collaborative group of researchers and practitioners from the Academy of Public Health Services (AÖGW), the Heinrich-Heine-Universität Düsseldorf (HHU) and the Local Health Authority Düsseldorf (LHA)—includes the following experts: Laura Arnold (AÖGW, project lead and corresponding author), Simon Bimczok (AÖGW), Hannah Schütt (AÖGW), Dagmar Starke (AÖGW), Nico Dragano (HHU), Delbar Dilmaghani (HHU), Annika Höhmann (HHU), Simon Götz (HHU), Simone Weyers (HHU), Ravina Ambalavanar (LHA), Anke Kietzmann (LHA), Andrea Melville-Drewes (LHA), Guido Schenuit (LHA), Trudpert Schoner (LHA). and Michael Schäfer (Freelancer).

**Additional information**

**Informed consent statement**

Written Informed consent was obtained from all subjects involved in the qualitative study.

## Author Contributions

**Conceptualization:** Laura Arnold.

**Data curation:** Laura Arnold.

**Formal analysis:** Laura Arnold, Simon Bimczok.

**Funding acquisition:** Laura Arnold, Dagmar Starke.

**Investigation:** Laura Arnold, Simon Bimczok.

**Methodology:** Laura Arnold, Simon Bimczok.

**Project administration:** Laura Arnold.

**Supervision:** Laura Arnold, Timo Clemens, Helmut Brand, Dagmar Starke.

**Validation:** Laura Arnold, Simon Bimczok.

**Visualization:** Laura Arnold, Simon Bimczok.

**Writing – original draft:** Laura Arnold, Simon Bimczok.

**Writing – review & editing:** Laura Arnold, Simon Bimczok, Timo Clemens, Helmut Brand, Dagmar Starke.

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
