## [Decision Letter · Decision Letter 0]

19 Dec 2023

PONE-D-23-29269Implementing evidence ecosystems in the public health service: Development of a seven-step framework for designing tailored training programsPLOS ONE

Dear Dr. Arnold,

Thank you for submitting your manuscript to PLOS ONE. After careful consideration, we feel that it has merit but does not fully meet PLOS ONE’s publication criteria as it currently stands. Therefore, we invite you to submit a revised version of the manuscript that addresses the points raised during the review process.

We look forward to receiving your revised manuscript.

Kind regards,

Kuo-Cherh Huang

Academic Editor

PLOS ONE

4. In the online submission form, you indicated that [The transcripts of the qualitative interviews are not available in full text for privacy reasons. An overview of the coding tree underlying the scoping review and a tabular summary of the included studies can be found in the appendix. The authors will be pleased to provide further results upon request. The learning objectives catalog and the associated handbook will be published as open access after completion of the evaluation of the pilot phase.].

5. One of the noted authors is a group or consortium [EvidenzÖGD study consortium]. In addition to naming the author group, please list the individual authors and affiliations within this group in the acknowledgments section of your manuscript. Please also indicate clearly a lead author for this group along with a contact email address.

Additional Editor Comments:

Dear Dr. Arnold,

Thank you for submitting your revised manuscript to PLoS ONE. Although your paper is timely and interesting, it is evident that your work could be improved further by addressing the expertly comments from both reviewers. I would like to suggest that you scrutinize your manuscript closely since there are several writing anomalies as assiduously pointed out by both reviewers. Thank you.

Kuo-Cherh Huang

Reviewers' comments:

Reviewer's Responses to Questions

**Comments to the Author**

1. Is the manuscript technically sound, and do the data support the conclusions?

Reviewer #1: Yes

Reviewer #2: Yes

2. Has the statistical analysis been performed appropriately and rigorously? 

Reviewer #1: Yes

Reviewer #2: N/A

3. Have the authors made all data underlying the findings in their manuscript fully available?

Reviewer #1: Yes

Reviewer #2: No

4. Is the manuscript presented in an intelligible fashion and written in standard English?

Reviewer #1: Yes

Reviewer #2: Yes

5. Review Comments to the Author

Reviewer #1: Extremely well written and easy to follow, even though portions on the paper are dense. I don't have any substantive suggestions. There appears to be an error in the reported number of records found on line 313. Additionally, the authors should consider providing examples, if possible, that demonstrate how this toolbox can be easily adapted and used across government structures and jurisdictional levels. Is that a limitation?

Reviewer #2: I had the opportunity to review this insightful and novel manuscript that focuses on the development of a seven-step framework for designing tailored training programs in public health. It is evident that the paper contributes significantly to the field, particularly due to its timeliness, the innovative approach and well-structured content. Below, I share my thoughts, suggestions, and recommendations that I believe could enhance the value and clarity of the final manuscript.

Major Comments

a. Introduction

Expansion on 'Judicious Application': The concept of 'judicious application' of scientific knowledge is pivotal. An elaboration on this would strengthen the introduction.

b. Aim and Framework Description

Distinguishing Between Framework and Toolbox: The current overlap between the framework and toolbox in the description needs clarification. Clear distinctions would enhance the reader's understanding.

c. Methods

Clarification of 'Stepwise Iterative' Method: The iterative aspect of the method is unclear. Expanding on this, along with detailed information about key informants and the involvement process in steps 1 and 3, would add clarity.

Ethical Issues and Data Triangulation: Suggesting a separate section for ethical issues and an expansion on data triangulation that can provide more depth to the methodology.

d. Results

Presentation of Participant Characteristics: Consider presenting participant characteristics in a table format for clarity and ease of understanding.

Criteria for Qualification Models: Clarification on the criteria used in the scoping review to identify qualification models is needed.

Additional Discussions on Toolbox: Discussion on the specificity of the toolbox in the context of public health discipline and the geographical specifics of Germany would be beneficial.

Minor Comments

Figure Corrections: There appears to be a mismatch in the naming and ordering of figures in the text. This needs correction for accuracy and consistency.

6. PLOS authors have the option to publish the peer review history of their article (what does this mean?). If published, this will include your full peer review and any attached files.

Reviewer #1: No

Reviewer #2: No

---

## [Author Response · Author response to Decision Letter 0]

12 Mar 2024

Response to reviewers

Reviewer #1: 

Reviewer #1, comment 1, general: Extremely well written and easy to follow, even though portions on the paper are dense. I don't have any substantive suggestions. 

Response: We would like to express our sincere gratitude for your constructive review of our manuscript. Your comments have been very helpful to improve the clarity, accuracy, and overall quality of our work.

Reviewer #1, comment 1, (a) There appears to be an error in the reported number of records found on line 313. 

Response: Thank you for pointing out the typographical error on line 313. The figure reference initially given as '1,1706' should have been '1,706'. We have corrected this error in the revised manuscript. 

Reviewer #1, comment 1, (b) Additionally, the authors should consider providing examples, if possible, that demonstrate how this toolbox can be easily adapted and used across government structures and jurisdictional levels. Is that a limitation?

Response: Thank you for raising that question. While we acknowledge the potential benefit of providing detailed examples of the toolbox's application, we regard this to be beyond the scope of this manuscript. We agree that providing examples of its usability is as such very valuable and we are currently working on a dedicated outlet to provide a detailed and comprehensive example of the toolbox's application. In this application example, we illustrate on how the toolbox was applied within the diverse administrative structures and levels of responsibility in the city of Düsseldorf, Germany by developing a training program based on a trainee-rotational concept. We are confident that this upcoming manuscript will be a valuable addition to this manuscript focusing on the development of the toolbox itself. We added references to this upcoming application example of the toolkit in several stages of the manuscript, for example in chapter 3.4 and chapter 4. While we agree very much about the added benefit of providing an application example, we would therefore suggest not adding detailed examples to the presented manuscript. We are, however, very grateful for your suggestion, as it encourages us in our work on the upcoming manuscript!

We do not perceive the adaptability of the toolbox to various governance structures and levels of responsibility as a limitation. The toolbox was intentionally designed with a generic framework to enable municipalities to tailor it to their specific regional structures. Within the scope of the superior EvidenzÖGD project, the toolbox has already been successfully employed to develop a customized training program tailored specifically to the municipal context in Düsseldorf: link. In this regard, we developed a competence-based catalogue of learning objectives and an accompanying handbook, both of which will be made publicly available later this year. Building on the valuable experience gained from this piloting phase, the toolbox will be used in various other municipalities in the coming year to foster local evidence ecosystems. We are very pleased, that two universities and three local health authorities have already expressed their interest in utilizing the toolbox to develop a tailored training program. 

We regard the generic framework would also be suitable for the development of tailored training programs at the state or even national level, assuming an appropriate partnership between public health and science. The generic toolbox was developed for the design of customized training offers to strengthen local evidence ecosystems. To this end, it is essential that partners from public health practice and public health science work together on an equal footing. In our opinion, further research is needed to examine the applicability of the toolbox at federal or state level in terms of practicability. Therefore, we added the following passage to the strengths and limitations section to our manuscript:

While our study explicitly focused on enhancing knowledge translation and exchange within the decentralized governance structure of the public health system in Germany, we recognize the potential limitations of the toolbox's applicability to systems with different structures. Our interviews and scoping review are tailored to the specific needs of the German context and therefore focus on five selected European countries, which we consider a strength. However, we intentionally designed the toolbox with a generic framework to enable adaptation to various regions and contexts. We acknowledge the potential utility and necessity of this adaptability in designing customized training programs for different systems. The toolbox's transferability to other geographical and administrative contexts is currently being explored in the ongoing EvidenzÖGD study. The extent to which the toolbox can be used effectively in diverse administrative systems and settings without decentralized governance structure must be examined in the future.

Reviewer #2: 

Reviewer #2, comment 1, general: I had the opportunity to review this insightful and novel manuscript that focuses on the development of a seven-step framework for designing tailored training programs in public health. It is evident that the paper contributes significantly to the field, particularly due to its timeliness, the innovative approach and well-structured content. Below, I share my thoughts, suggestions, and recommendations that I believe could enhance the value and clarity of the final manuscript.

Response: We greatly appreciate the time you took to review our article and all the valuable comments you provided us with. We regard your comments to be instrumental to improve the clarity, accuracy, and overall quality of our work. In the following, we briefly refer to each of your comments separately.

Reviewer #2, comment 2, Introduction: Expansion on 'Judicious Application': The concept of 'judicious application' of scientific knowledge is pivotal. An elaboration on this would strengthen the introduction.

Response: We have incorporated an expansion on the concept of "judicious application" in the introduction section, emphasizing its significance at the municipal level. Your guidance is greatly appreciated. We added the following details regarding the concept of "judicious application" in the introduction section: 

Ensuring effective resource utilization and successful intervention implementation requires not only access to scientific knowledge but also its judicious application (Brownson et al. 2009), including the thoughtful integration of evidence into practice, consideration of the specific context, continuous evaluation and adaptation of measures and strategies, and continuous stakeholder engagement to ensure ethical, effective, and sustainable outcomes. A judicious approach has the potential to provide decision-makers with better access to information on effective strategies, thereby, optimizing the allocation of limited resources and increasing confidence in the successful implementation of different intervention options (Kneale et al. 2019). Especially during periods of financial constraints, considering carefully the intricacies of practical implementation within public health decision-making processes safeguards the well-being of the population. This is particularly relevant for the municipal level, as LHAs possess an encompassing understanding of local conditions and contexts predestining them to ensure EIDM processes at the local level (Starke und Arnold 2021; Kuhn und Wildner 2020; Szagun et al. 2016). 

Reviewer #2, comment 3, Aim and Framework Description: Distinguishing Between Framework and Toolbox: The current overlap between the framework and toolbox in the description needs clarification. Clear distinctions would enhance the reader's understanding.

Response: We appreciate your valuable feedback. To emphasize the difference more clearly, we have simplified the use of both terms within the manuscript. 

To clarify the difference, we have adapted the abstract as follows:

To address this issue, we developed a framework incorporating a toolbox to conceptualize training programs designed to qualify public health professionals for working at the interface between academia and practice. We conducted a scoping review of training programs, key-informant interviews with public health experts, and a multi-professional stakeholder workshop and triangulated their output. The resulting toolbox consists of four core elements, encompassing 15 parameters: (1) content-related aspects, (2) context-related aspects, (3) aspects relevant for determining the training format, and (4) aspects relevant for consolidation and further development. Guiding questions with examples supports the application of the toolbox. Additionally, we introduced a how-to-use guidance to streamline the creation of new training programs, fostering knowledge transfer

Furthermore, we added a definition of both terms in the method section:

In our study, the terms "framework" and "toolbox" are closely interlinked: The “framework” refers to the overarching theoretical construct guiding the development of tailored training programs. In contrast, the “toolbox” consists of a comprehensive collection of essential elements and parameters, identified during our research. It provides researchers and practitioners with customized guiding questions to identify and discuss all components relevant to their field. Together with its accompanying short application guide, the toolbox is a pivotal aspect of the framework, integral to its function and effectiveness.

 We also emphasized the difference in the discussion section: 

The toolbox presented is deliberately generic so that it can be applied to different contexts. To this end, the accompanying guiding questions are intended to be supportive to determine the relevant focus and content. While the toolbox can be used standalone by systematically answering all questions in sequence, we believe the toolbox to be most useful in an iterative deliberative process, as recommended by experts in the field (Batt et al. 2021). To facilitate this, we propose a seven-step process, visually depicted in Figure 3, with a brief explanation of each step outlined below.

 We also highlighted the integration of the toolbox in the how-to-use guide:

During the development phase (step 3), the core team uses the toolbox to reach a consensus on the fundamental elements of the training program. An iterative process is recommended refining content- as well as context-related aspects by systematically assessing the provided guiding questions (see Table 2 and 3) in successive workshops conducted by the core team. To ensure continuous exchange within the core team, a predetermined schedule should be agreed upon in the initial meeting. Once program objectives and content are specified, the program format, including duration and density, professional credentialing requirements, and desired recruitment and selection criteria, can be defined (see Table 43 and 5). 

We thank you for your valuable input and are committed to making the necessary clarifications to improve the clarity and coherence of our manuscript.

Reviewer #2, comment 4, Methods: (a) Clarification of 'Stepwise Iterative' Method: The iterative aspect of the method is unclear. Expanding on this, along with detailed information about key informants and the involvement process in steps 1 and 3, would add clarity.

Response: (a) Thank you for pointing out that the iterative aspect of our approach was unclear in the current manuscript. For clarification, we have included the following information in the method section of the revised manuscript. 

Through an iterative process, data from steps one to three were subjected to reflexive analysis, where the results were compared and synthesized. These consolidated outcomes, summarized by two researchers (LA, SB), were subsequently presented to the project consortium members for critical assessment. The feedback received informed further revisions. This iterative cycle continued until a consensus was achieved. As a result of this cyclical process, a comprehensive toolbox was developed, aimed at facilitating the creation of customized training programs for public health professionals who work at the interface.

We also included information on the selection, roles, and contributions of the key informants in steps 1 and 3 in the manuscript:

Chapter 3.1 Qualitative interview study:

Key informants were carefully chosen based on their expertise in public health academia and practical experience in the PHS. We aimed to achieve a diverse representation, considering a variety of professional backgrounds and levels of experience.

Chapter 3.3 Stakeholder workshop:

The participant pool was intentionally diverse, extending beyond the key informants from the qualitative interviews. It included representatives from various public health organizations, public health practitioners from PHS on local, state and federal level, academic researchers from different universities and research centers, and young professionals. The selection was guided by the intent to include individuals who shared professional and communication skills but offered a range of opinions, to encourage dynamic and critical discussions.

Reviewer #2, comment 4, Methods: (b) Ethical Issues and Data Triangulation: Suggesting a separate section for ethical issues and an expansion on data triangulation that can provide more depth to the methodology.

Response: We separated the ethical issues and the data triangulation by introducing a new chapter “3.5 Ethics and reflexive statement”. Furthermore, we have expanded the ethical aspects to include a reflexivity statement: 

Throughout the qualitative interviews and the stakeholder workshop, our research team maintained active engagement with both the research process and the participants. In line with the COREQ guidelines, we acknowledge that personal biases were not entirely avoidable. This recognition led us to adopt a reflexive stance throughout the research, where we remained aware of our potential biases and their potential impact on data collection and analysis. To mitigate the influence of these biases, we implemented several measures: We critically examined our assumptions, fostering a culture of self-reflection and open discussion. Regular monthly meetings with the project consortium were instrumental in this regard, providing an interdisciplinary platform for evaluating research progress and perspectives.

Our research team brought together a diverse range of expertise, including epidemiology (LA, DD), medicine (DD, HB), public health (LA, SB, HB, TC, SG, DS), sociology (SW, DS), and social sciences (LA, DS). All authors possessed experience in conducting qualitative research projects. This diversity in professional and academic backgrounds played a pivotal role in shaping the research process. During the triangulation and synthesis phase, the multifaceted perspectives within our team proved to be particularly valuable. Each member's insights and interpretations, shaped by their unique disciplinary lenses, were critical in the development of the final toolbox.

In the section '3.4 Triangulation and Synthesis,' we have expanded on our methodology to provide a more detailed account of the triangulation and synthesis process, including the iterative cycle of reflection and analysis undertaken by our team: 

Through an iterative process, data from steps one to three underwent reflexive analysis, results were compared, and synthesized. These consolidated outcomes were summarized by two researchers (LA, SB) and then presented to the members of the project consortium for critical assessment. The feedback obtained guided subsequent revisions. This iteration continued until a saturation point was reached. As a result of this cyclical process, a comprehensive toolbox was developed to facilitate the creation of customized training programs for public health professionals working at the intersection of science and practice.

We believe these amendments provide greater depth to our methodology and would like to thank you for your suggestion.

Reviewer #2, comment 5, Results: (a) Presentation of Participant Characteristics: Consider presenting participant characteristics in a table format for clarity and ease of understanding.

Response: Thank you for your suggestion. We have included a table in the results section, which outlines the key characteristics of the participants.

Reviewer #2, comment 5, Results: (b) Criteria for Qualification Models: Clarification on the criteria used in the scoping review to identify qualification models is needed.

Response: We have addressed this point in the revised manuscript, with details now incorporated in both the methods and results sections. 

In the methods section, we describe the criteria used for selecting the models, which were based on their compatibility with the PHS workforce in Germany and their practice-oriented nature:

The selection of these training programs was a collaborative decision within the consortium, informed by the outcomes of both the qualitative interview study and the scoping review. This selection process was guided by a dual focus: the programs chosen were those deemed most compatible with the operational environment of the PHS workforce, and simultaneously, those that were practice-oriented, facilitating effective work at the interface between scientific research and practical application.

In the results section, we connect these criteria to the specific models identified as most suitable: 

Within the scoping review, we identified various qualification models, with five being particularly suitable for facilitating evidence use and knowledge transfer within the local PHS in Germany. These models include trainee programs, PhD programs, rotational concepts, continuing professional development (CPD) courses that integrate both academia and practice, and PHS-related modules in academic degree programs. The selection of these models was based on a collaborative endeavor within the consortium, deeply informed by insights from the qualitative interviews, the scoping review, and the stakeholder workshop. Our criteria for choosing these training programs were twofold: First, we concentrated on programs compatible with the operational environment of the PHS workforce in Germany, ensuring alignment with the specific requirements and challenges faced by professionals. Second, we prioritized programs that were practice-oriented, designed to effectively bridge the gap between scientific research and practical application in public health service. This dual-focused selection process was aimed at identifying programs that meet the contextual needs of the PHS workforce while also enhancing knowledge transfer and application at the interface of academia and practice.

Reviewer #2, comment 5, Results: (c) Additional Discussions on Toolbox: Discussion on the specificity of the toolbox in the context of public health discipline and the geographical specifics of Germany would be beneficial.

Response: Thank you for your insightful comment. While our approach intentionally centered on the German context to address the unique aspects of its federal structure, we acknowledge that this focus could represent a limitation. In response to your suggestion, we have added the following paragraph to the strengths and limitation section to provide further clarity:

While our study explicitly focused on enhancing knowledge translation and exchange within the decentralized public health system governance structure of Germany, we recognize the potential limitations of the toolbox's applicability to systems with different structures. Our interviews and scoping review had a European focus, tailored to the unique needs of the German context, which we consider a strength. However, we intentionally designed the toolbox with a generic framework to enable adaptation to various regions and contexts. We acknowledge the potential utility and necessity of this adaptability in designing customized training programs for different systems. The toolbox's transferability to other geographical and administrative contexts is currently being explored in the ongoing EvidenzÖGD study (Arnold et al., in preparation). Further research is needed to assess the extent to which the toolbox can effectively serve in diverse administrative systems and settings without a decentralized governance structure.

Reviewer #2, comment 6, Figure Corrections: There appears to be a mismatch in the naming and ordering of figures in the text. This needs correction for accuracy and consistency.

Response: Thank you for pointing out the mismatch. We double checked the naming and ordering of all figures and tables and corrected them if necessary in the revised manuscript.

---

## [Editor Report · Decision Letter 1]

18 Mar 2024

Implementing evidence ecosystems in the public health service: Development of a framework for designing tailored training programs

PONE-D-23-29269R1

Dear Dr. Arnold,

We’re pleased to inform you that your manuscript has been judged scientifically suitable for publication and will be formally accepted for publication once it meets all outstanding technical requirements.

Kind regards,

Kuo-Cherh Huang

Academic Editor

PLOS ONE
---

## [Editor Report · Acceptance letter]

27 Mar 2024

PONE-D-23-29269R1 

PLOS ONE

Dear Dr. Arnold, 

I'm pleased to inform you that your manuscript has been deemed suitable for publication in PLOS ONE. Congratulations! Your manuscript is now being handed over to our production team.

Kind regards, 

on behalf of

Dr. Kuo-Cherh Huang 

Academic Editor

PLOS ONE